# Effectiveness and cost-effectiveness of personalised dietary advice aiming at increasing protein intake on physical functioning in community-dwelling older adults with lower habitual protein intake: rationale and design of the PROMISS randomised controlled trial

Ilse Reinders [1], Hanneke A H Wijnhoven,[1] Satu K Jyväkorpi,[2,3] Merja H Suominen,[2,3] Riikka Niskanen,[2,3] Judith E Bosmans,[1] Ingeborg A Brouwer,[1] Kristien S Fluitman,[4,5] Michel C A Klein,[6] Lothar D Kuijper,[1] Laura M van der Lubbe,[6] Margreet R Olthof,[1] Kaisu H Pitkälä,[2,3] Rachel Vijlbrief,[1] Marjolein Visser[1]

For numbered affiliations see end of article.

**Correspondence to**
Dr Ilse Reinders;
ilse.reinders@vu.nl

## ABSTRACT

**Introduction** Short-term metabolic and observational studies suggest that protein intake above the recommended dietary allowance of 0.83 g/kg body weight (BW)/day may support preservation of lean body mass and physical function in old age, but evidence from randomised controlled trials is inconclusive.

**Methods and analysis** The PRevention Of Malnutrition In Senior Subjects in the EU (PROMISS) trial examines the effect of personalised dietary advice aiming at increasing protein intake with or without advice regarding timing of protein intake to close proximity of usual physical activity, on change in physical functioning after 6 months among community-dwelling older adults (≥65 years) with a habitual protein intake of <1.0 g/kg adjusted (a)BW/day. Participants (n=264) will be recruited in Finland and the Netherlands, and will be randomised into three groups; two intervention groups and one control group. Intervention group 1 (n=88) receives personalised dietary advice and protein-enriched food products in order to increase their protein intake to at least 1.2 g/kg aBW/day. Intervention group 2 (n=88) receives the same advice as described for intervention group 1, and in addition advice to consume 7.5–10 g protein through protein-(en)rich(ed) foods within half an hour after performing usual physical activity. The control group (n=88) receives no intervention. All participants will be invited to attend lectures not related to health. The primary outcome is a 6-month change in physical functioning measured by change in walk time using a 400 m walk test. Secondary outcomes are: 6-month change in the Short Physical Performance Battery score, muscle strength, body composition, self-reported mobility limitations, quality of life, incidence of frailty, incidence of sarcopenia risk and incidence of malnutrition. We also investigate cost-effectiveness by change in healthcare costs.

## Strengths and limitations of this study

► This large randomised controlled trial addresses a key question whether dietary advice to increase protein intake to ≥1.2 g/kg adjusted body weight (aBW)/day is beneficial for physical functioning in community-dwelling older adults.

► This trial will also examine the additional effect of the timing of protein intake in close proximity of usual physical activity on change in physical functioning.

► Participants included had a habitual protein intake of <1.0 g/kg aBW/day.

► The lack of blinding of the study participants and nutritionists who also collect data on all outcome measures is a limitation of the study design.

► Another limitation of this study is that the biological value of the total protein intake (ie, type of amino acids) is unknown.

**Discussion** The PROMISS trial will provide evidence whether increasing protein intake, and additionally optimising the timing of protein intake, has a positive effect on the course of physical functioning after 6 months among community-dwelling older adults with a habitual protein intake of <1.0 g/kg aBW/day.

**Ethics and dissemination** The study has been approved by the Ethics Committee of the Helsinki University Central Hospital, Finland (ID of the approval: HUS/1530/2018) and The Medical Ethical Committee of the Amsterdam UMC, location VUmc, Amsterdam, the Netherlands (ID of the approval: 2018.399). All participants provided written informed consent prior to being enrolled onto the study. Results will be submitted for publication in peer-reviewed

journals and will be made available to stakeholders (ie, older adults, healthcare professionals and industry).

**Trial registration number** ClinicalTrials.gov Registry (NCT03712306).

## INTRODUCTION

There is an ongoing debate on whether or not older adults should be recommended a protein intake above the current recommended daily allowance (RDA) established by the European Food Safety Authority (EFSA) of 0.83 g/kg body weight (BW)/day for adults.[1] International panels of geriatricians, nutritional experts and scientists have proposed at least 1.0–1.2 g protein/kg BW/day for healthy older adults in order to maintain and regain muscle mass, strength and function.[2 3]

The proposed increase of the RDA for older adults is merely based on results from short-term metabolic and epidemiological studies. Several metabolic studies showed that older adults (≥65 years) have a lower muscle protein synthesis (MPS) following protein intake compared with younger adults,[4–6] and that higher protein intake enhances MPS in older adults when compared with lower protein intake (1.2 g/kg BW/day vs 0.8 g/kg BW/day,[7] or ≥30 g/day vs 15 g/day[8]). In addition, the anabolic threshold (ie, optimal dose of dietary protein in a meal that stimulates MPS) is 70% higher in older compared with younger adults.[5] Epidemiological studies have shown that higher dietary protein intake in older adults, defined as >0.9 g/kg BW/day[9] or >1.0 g/kg BW/day[10–12] is associated with lower risk of weight loss,[11] better disability trajectories,[12] less loss of lean mass[9] or lower risk of developing functional impairments.[10]

Despite the evidence from metabolic and epidemiological studies, causal evidence to support beneficial effects of protein intake at or above 1.0 g/kg BW/day based on randomised controlled trials (RCTs) is not conclusive. One systematic review showed no beneficial effect of increasing protein intake on lean body mass, muscle cross-sectional area, muscle strength or physical performance.[13] Of the 36 studies included in the systematic review, 26 studies presented mean habitual protein intake of the study participants which ranged between 0.78 and 1.5 g/kg BW/day, with only one study below the protein RDA of 0.8 g/kg BW/day.[13] The relatively high mean habitual protein intake may explain the absence of a beneficial effect of additional protein. Another explanation may be that the amount of protein provided might not have been sufficient to augment MPS. That is, a protein intake of 25–30 g is required to stimulate MPS and maintain muscle mass,[14 15] though the amounts provided varied between 10 g/day (3 days/week) and total intake of 125 g/day or were not reported. Of the trials published after the systematic review, Park et al showed that intake of 1.5 g/kg BW/day for 12 weeks resulted in higher muscle mass and improved gait speed compared with intake of 0.8 g/kg BW/day in undernourished prefrail and frail older adults.[16] Ten Haaf et al showed a positive effect of increasing protein intake for 12 weeks on lean body mass in active older adults with a habitual protein intake of <1.0 g/kg BW/day.[17] Beelen et al found no effects of protein supplementation on physical performance among older adults after hospital discharge,[18] however, baseline protein intake was already 1.0 g/kg BW/day in the control group and 1.5 g/kg BW/day in the intervention group. Finally, Bhasin et al showed no beneficial effects other than a decrease in fat mass after a controlled diet with 1.3 g/kg BW/day of protein for 6 months compared with a control diet consisting of 0.8 g protein/kg BW/day[19] among functionally limited community-dwelling men aged ≥65 years. However, mean body mass index (BMI) of the participants was quite high (30.3 kg/m$^2$), which may have resulted in an overestimation of baseline protein requirements. Based on inconsistent findings, more RCTs in older adults with lower habitual protein intake are needed to determine the potential effect of increasing protein intake on physical functioning outcomes.

Previous studies among older adults showed that protein supplementation in combination with resistance exercise has more beneficial effects on body composition, muscle strength and physical function compared with resistance exercise alone.[20–26] The underlying hypothesis is that protein supplementation augments the adaptive response of skeletal muscle to resistance exercise. In addition, there is evidence that timing of protein intake in close proximity of physical activity stimulated MPS to greater extent than when timed at other hours during the day.[27] To our knowledge, there are no RCTs investigating the effect of timing protein intake in close proximity of physical activities on physical functioning.

The PRevention Of Malnutrition In Senior Subjects in the EU (PROMISS) trial is designed to fill in some of the current knowledge gaps on the optimal amount of dietary protein in older community-dwelling adults and timing of protein intake in relation to physical activity. Its primary objective is to examine the effectiveness of personalised dietary advice aiming at increasing protein intake to at least 1.2 g/kg adjusted (a)BW/day on change in physical functioning after 6 months measured by change in walk time using a 400 m walk test among community-dwelling older adults with a habitual protein intake of <1.0 g/kg aBW/day. Additionally, it examines the combined effect of personalised dietary advice aiming at increasing protein intake to at least 1.2 g/kg aBW/day and advice aiming at optimising the timing of protein intake in close proximity of usual physical activity. The secondary objectives are to examine the effectiveness of personalised dietary advice aiming at increasing protein intake to at least 1.2 g/kg adjusted on 6-month changes in physical functioning measured by the Short Physical Performance Battery (SPPB) score, muscle strength, body composition, self-reported mobility limitations, quality of life (QoL), incidence of frailty, incidence of sarcopenia risk, and incidence of malnutrition and change in healthcare costs.

In three ancillary studies the following additional objectives are addressed; (1) the effect of using persuasive

technology on adherence to personalised dietary advice aiming at increasing protein to at least 1.2 g/kg aBW/ day, (2) the effect of personalised dietary advice aiming at increasing protein intake to at least 1.2 g/kg aBW/day on the oral and gut microbiota composition, and (3) the effect of personalised dietary advice aiming at increasing protein intake to at least 1.2 g/kg aBW/day on central neural responses to food-cues in brain areas of interest.

## METHODS AND ANALYSIS
### Study design
The PROMISS trial is a multicentre RCT designed to examine the effectiveness of personalised dietary advice aiming at increasing protein intake and advice on optimising the timing of protein intake in close proximity of usual physical activity on change in physical functioning after 6 months. Participants will be randomised into three groups: one control group (no intervention); intervention group 1 receiving personalised dietary advice aiming at increasing protein intake to at least 1.2 g/kg aBW/day; and intervention group 2 receiving personalised dietary advice aiming at increasing protein intake to at least 1.2 g/kg aBW/day, including personalised advice to optimise the timing of protein intake in close proximity of usual physical activity.

### Eligibility criteria
The eligibility criteria are proposed to include a study group of community-dwelling older adults (65+ years) with a habitual protein intake <1.0 g/kg aBW/day. Inclusion and exclusion criteria are listed in box 1, and some are described in more detail below.

Older adults with a BMI of <18.5 kg/m² will be excluded, because these participants are likely to be undernourished[28] and should preferably receive general nutritional care that is not provided in this trial. Those with a BMI of >32.0 kg/m² will be also excluded, because a high BMI (>30.0 kg/m²) is associated with poorer physical function[29] and disability[30] in old age, and intentional weight loss by lifestyle interventions leads to a reduced mortality risk.[31] In light of this evidence, older adults with a BMI >32.0 kg/m² should preferably be advised to lose weight, which is not the aim of the present study and may interfere with the study objective. Because participants of intervention group 2 will be advised to consume protein-rich foods in close proximity of usual physical activity, older adults who are bedridden, wheelchair users or do not go outside will be excluded from the trial. Older adults with a diagnosis of severe kidney disease (ie, treatment of a nephrologist and/or protein-restricted diet, self-reported) will also be excluded as they should be advised to limit their protein intake.[2 32–34] Older adults with a low cognitive status (Mini-Mental State Examination (MMSE) score ≤20[35]) will be excluded, as participants should be able to understand and follow dietary advice if randomised to one of the intervention groups.

### Calculation of protein intake using aBW
To calculate habitual protein intake in g/kg aBW/ day for all (potential) participants and recommended protein intake (participants in the two intervention groups), we apply aBW depending on participants' age and BMI. We use aBW because underweight persons require extra protein to build muscle tissue, while in overweight persons, much 'extra weight' is adipose tissue. Protein intake in g/kg aBW/day is based on self-reported BW during screening and afterwards based on measured BW during the baseline assessment, which is further used throughout the study. For those with a BMI >25.0–32.0 kg/m² (age ≤70 years) or >27.0–32.0 kg/m² (age >70 years), we apply aBW corresponding to a BMI of, respectively, 25.0 or 27.0 kg/m². For those with a BMI >18.5–<22.0 kg/m² (age >70 years), we apply aBW corresponding to a BMI of, respectively, 18.5 or 22.0 kg/m².[36]

For the recommended protein intake, we apply aBW which is based on baseline measured BW.

## Recruitment and screening

Two hundred and sixty-four community-dwelling adults aged 65 years and older will be recruited at two study sites (metropolitan area of Finland including Helsinki, Espoo, Vantaa, Kauniainen, and Amsterdam, the Netherlands). The recruitment strategy includes mass mailing using addresses obtained from a random sample of the Finnish Population Registry (in Finland only), newspaper advertisements, media coverage, lectures, oral presentations to the target group, informing professionals working with older adults and flyers which will be distributed at locations where many community-dwelling older adults visit.

Older adults who are interested in participating will be asked to contact the local PROMISS research team (by phone or by email). Thereafter, screening by phone takes place, only when verbal informed consent is given, in which the majority of the eligibility criteria will be assessed along with an explanation of the study. Only those with a lower habitual protein intake (<1.0 g/kg aBW/day) will be invited for the first clinic visit. Assessment of habitual protein intake will be estimated in two steps: (1) initial screening by phone; (2) a full dietary assessment based on a combination of three food diaries and three 24-hour dietary recalls to confirm lower habitual protein intake. Step 1, the initial screening is performed by phone using the Protein Screener 55+ (Pro55+, available for use in English, Finnish and Dutch: see www.proteinscreener. nl/#/). This screening tool was specifically developed and validated for this purpose.[37] The screening results in a probability score (0%–100%) of having a protein intake below 1.0 g/kg aBW/day. At a probability of >30%, sensitivity and specificity are optimally balanced.[37] In the PROMISS trial we select persons with a probability score varying between >15% (when initial response rates to recruitment strategies are low) and >30% (when initial response rates are high), for the second step of assessing habitual protein intake. Those who fulfil the eligibility criteria receive further information on the study and a food diary with a booklet with pictures of portion sizes by post to support the 24-hour dietary recalls. After a minimum of 1 week of consideration, the research staff contacts the older adults, and among those who are still willing to participate the full dietary assessment will take place (step 2). These potential participants will be asked to keep track of their dietary intake by filling out the provided food diary for 3 consecutive days (3 weekdays; or 2 weekdays and 1 weekend day). The booklet with pictures of portion sizes that they received earlier will help them accurately fill out the diary. Each day after, they will be called by a nutritionist to go through their food diary of the day before (24-hour dietary recall). Potential participants are asked whether these days are representative for their habitual diet. In case one of the 3 days is not representative, mean protein intake is based on 2 instead of 3 days. In case of more than 1 non-representative

day, the person will be excluded. The food intake data based on the 24-hour dietary recall will be entered into the program 'Fineli' for the Finnish data[38] and into the program 'Eetmeter' of the Dutch Nutrition Center using an extended version of the Dutch Food Composition Table of 2016 for the Dutch data[39] to calculate intake of macronutrients and micronutrients (vitamin D and vitamin $B_{12}$). Participants with an actual protein intake ≥1.0 g/kg aBW/day (based on self-reported BW) will be excluded.

Potential participants with a mean habitual protein intake <1.0 g/kg aBW/day (based on the three 24-hour dietary recalls) will be invited for the clinic visit, where final eligibility criteria will be assessed; MMSE >20, ability to walk 400 m within 15 min (the use of a cane is allowed, but without the use of a walker and no rest longer than 60 s), and BMI of ≥18.5 kg/m$^2$ and ≤32.0 kg/m$^2$ based on measured BW and body height. When all eligibility criteria are met, participants are included in the PROMISS trial.

## Randomisation, allocation and masking

Randomisation by means of a stratified block randomisation procedure will be performed by an independent statistician. Participants will be allocated in a 1:1:1 ratio to the three groups. The size of the randomisation blocks is three. Participants will be stratified according to their baseline habitual protein intake (<0.9 or 0.9–1.0 g/kg aBW/day) and sex to ensure homogeneous distribution of baseline habitual protein intake and sex in the three groups across the two recruitment sites, because there may be a different intervention effect by baseline habitual protein or sex. In case couples are eligible, we will allocate them to the same intervention group to limit interference between intervention groups. We will randomly select on which partner the randomisation for the intervention group is based. Any resulting unbalance in the number of subjects per treatment arm will be corrected in the randomisation of the next block. Due to the nature of the study, researchers, nutritionists and participants are not blinded to the study group.

## Study timeline

The first clinic visit starts with written informed consent, and when participants are eligible, the baseline assessment will be performed. The baseline assessment consists of questionnaires (frailty status, risk of sarcopenia, self-reported mobility limitations, QoL and healthcare costs) and measurements (physical function, muscle strength and body composition). See below 'primary and secondary outcomes' and 'other measures' for details on these assessments. An accelerometer will be attached to measure physical activity for 7 subsequent days.

After the baseline assessment, participants will be randomised to one of the three study groups done by the nutritionists and they will inform the participants in which group they are allocated to. Participants randomised to one of the two intervention groups will be invited for a consultation meeting at the clinic to receive their

personalise d dietary advice, and personalised advice on optimising the timing of protein intake in close proximity of usual physical activity (intervention group 2 only). This will take place within 2 weeks after the baseline assessment since the personalised advice needs to be composed by the nutritionist. The baseline assessment is considered the start of the study period for participants of the control group, while the consultation meeting is considered the start of the study period for participants of the intervention groups.

One week prior to the 3-month follow-up visit, dietary intake will be assessed again by means of a combination of three food diaries and three 24-hour dietary recalls. The 3-month follow-up visit will take place at the clinic and includes measurement of BW, assessment of self-reported mobility limitations, risk of sarcopenia, QoL, healthcare costs and the accelerometer will be attached to measure physical activity for 7 subsequent days.

One week prior to the 6-month follow-up visit (final measurement), dietary intake will again be assessed by means of a combination of three food diaries and three 24-hour dietary recalls, which allows us to determine compliance to the dietary advice. The 6-month follow-up visit at the clinic includes all measurements performed during the baseline visit and the accelerometer is attached again to measure physical activity for the next 7 days. Finally, among participants of the intervention groups only, several questions regarding the appreciation and adherence of the intervention and participants' intention to follow the dietary advice in the future will be asked in order to perform a process-evaluation. Figure 1 shows the study timeline and table 1 provides an overview of all measurements.

## Intervention

Participants in the intervention group 1 will receive a personalised dietary advice face-to-face by nutritionists dedicated to this study aiming at increasing their protein intake to at least 1.2 g/kg aBW/day (without increasing total daily energy intake), based on their habitual dietary characteristics (based on three 24-hour recalls), protein intake and measured BW assessed at baseline. Participants will be asked if they usually prepare the main meal; whether they eat the meal at a for example, community home; whether they consume ready-to-eat meals; whether they use meal services; and if they eat at family or friends' home. All their answers will be incorporated in the personalised dietary advice. Participants will receive written dietary advice accompanied by a verbal explanation from the nutritionist. Participants can contact the nutritionist at any time by mail or phone in case any question arises. The advice includes the use of regular protein-rich food products and protein-enriched food products provided by the research team, and will be based on personal dietary preferences. Protein-enriched food products that can be incorporated within the regular diet include protein bars, cereals, puddings, coconut water and whey powder, which will be freely provided and shipped to participants' home.

Those products can be incorporated in the dietary advice as they can make it easier to increase protein intake due to their high-protein content.

Participants receive guidelines on how to incorporate the protein-enriched food products within their diet. The dietary advice will also incorporate the advice to consume at least one daily meal consisting of ≥35 g protein, as studies have shown that this amount increases MPS in older adults.[40–42]

Participants in intervention group 2 will receive the same dietary advice as intervention group 1, and the personalised advice to consume at least 7.5–10 g protein through protein-(en)rich(ed) food products within half an hour after performing usual physical activity as this may enhance resistance exercise-induced MPS.[27] One RCT among older adults has shown that protein supplementation in combination with resistance exercise had beneficial effects on, for example, muscle mass and function, but no differences in effect were found between protein consumption pre-resistance versus post-resistance exercise.[43] We therefore recommend protein intake after physical activity as this is a uniform and more feasible advice compared with 'in close proximity of', and might also result in less stomach discomfort as compared with protein consumption prior to physical activity. Usual physical activity is defined as either physical exercise (eg, biking, swimming, tennis) or the most intensive activities of daily living when the participant does not engage in physical exercise (eg, gardening, housekeeping, doing groceries) for a minimum of 30 min. The advice is linked to most extensive or longest physical activity. Participants are instructed not to become more or less physically active but merely to shift their physical activity or protein intake moment.

During the intervention period, nutritionists will plan follow-up phone calls in consultation with the participants during week 2, week 4, week 8, week 16 and week 20 to ask if they have understood the advice and are able to adhere to the advice. In addition, any issues related to the use of protein-enriched food product can be discussed (intervention groups only). If necessary, changes in the dietary advice will be made, for example when weight change >2 kg has occurred (based on self-assessment). Participants allocated to the control group do not receive any intervention, but are contacted on similar time points as the intervention groups to ask how they are doing.

All participants are invited to a minimum of one organised lecture on non-health-related themes and other social events during the trial in order to stimulate their commitment to the trial. Separate lectures/events (with the same topic) are organised for the intervention groups and the control group to prevent interference between intervention arms. Participants can freely attend those lectures and all travel costs will be reimbursed.

## Compliance

We will collect dietary intake prior to the 3-month follow-up visit (by means of the combination of three food

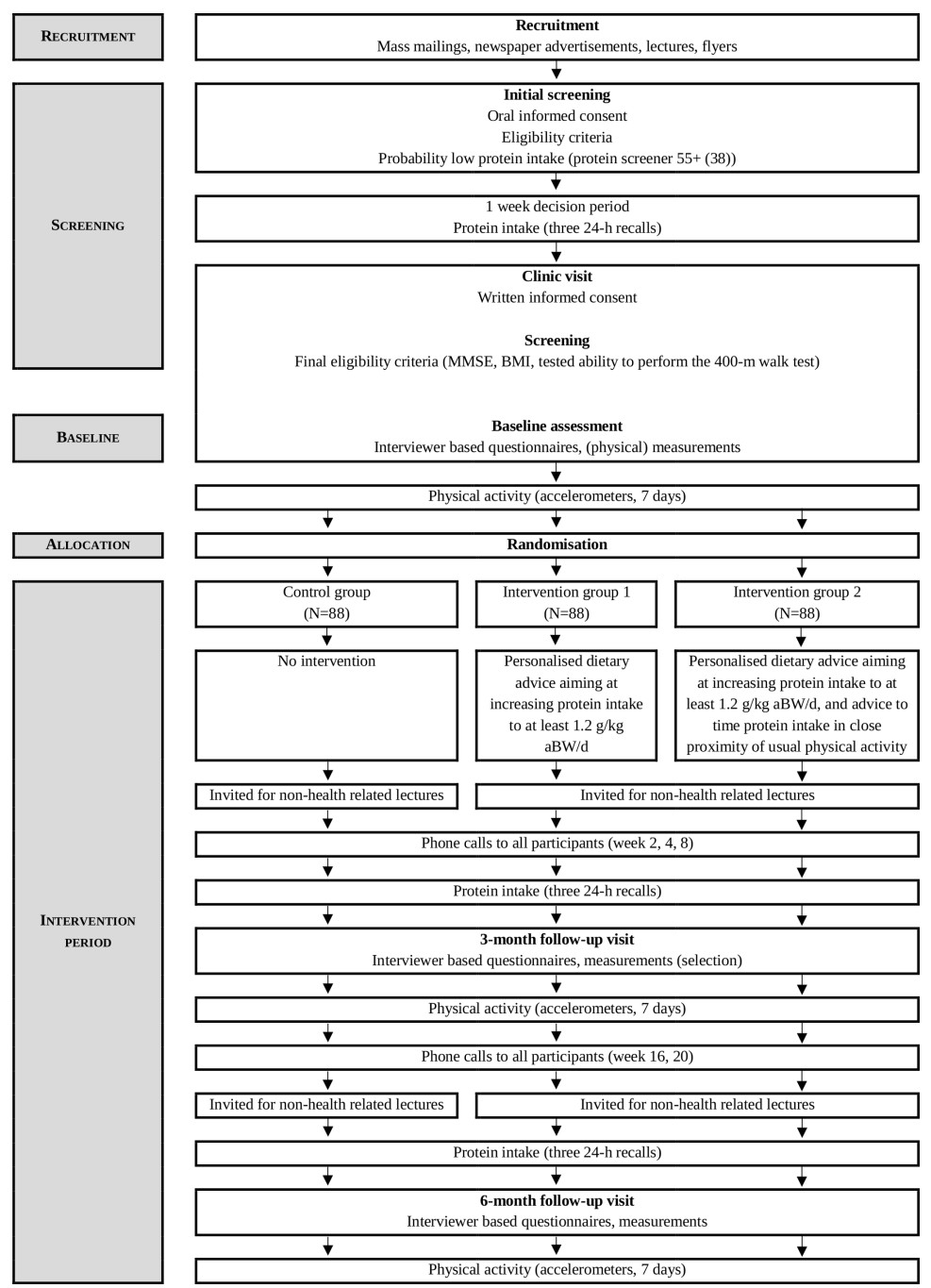

**Figure 1** Study timeline of the PROMISS trial. aBW, adjusted body weight; BMI, body mass index; MMSE, Mini-Mental State Examination; PROMISS, PRevention Of Malnutrition In Senior Subjects in the EU.

diaries and three 24-hour dietary recalls) to assess compliance to the dietary advice. This information allows the nutritionists to provide additional advice—if needed—for participants in the intervention groups, which will be provided during the 3-month follow-up visit. Dietary intake will again be assessed at follow-up and compliance to the dietary advice will be determined.

### Intervention fidelity

To ensure good adherence to the intervention protocols at both study sites, all personnel working for the trial have undergone extensive training. The nutritionists will follow written standardised operational procedures to develop and provide the personalised dietary advice (with or without additional advice to consume protein within half an hour after usual physical activity). Four times during the conduct of the trial, the nutritionists from one site will visit the other site to attend assessments, and potentially notice and correct differences in order to ensure identical execution of the trial at both sites. In addition, monthly Skype meetings will be held between all staff involved in the execution of the trial at both sites to solve any potential day-to-day issues in a standardised

**Table 1** Measurements and time of measurements in the PROMISS trial

| Topic | Specific variables | Screening (phone) | Screening visit (prior baseline) | Baseline visit | 3-month FU visit | 6-month FU visit |
|---|---|---|---|---|---|---|
| | | | | **Timeline** | | |
| Oral informed consent (phone) | | ✔ | | | | |
| Screening questionnaire (phone) | Sex, age, self-reported weight, height, eligibility criteria | ✔ | | | | |
| Protein intake | Pro55+ screening[37] (phone) | ✔ | | | | |
| Protein intake | Combination of three food diaries and three 24-hour dietary recalls | | ✔ | | ✔ | ✔ |
| Written informed consent | | | ✔ | | | |
| Cognitive function | MMSE[35] | | ✔ | | | |
| Physical functioning | 400 m walk test[44 45] | | ✔ | | | ✔ |
| Antropometrics | Measured body height | | ✔ | | | |
| Antropometrics | Measured body weight | | ✔ | | ✔ | ✔ |
| Demographic | Education, household | | | ✔ | | |
| General characteristics | Perceived health, smoking status | | | ✔ | | ✔ |
| Body composition | Bioelectrical impedance | | | ✔ | | ✔ |
| Body composition | Air displacement plethysmography (Dutch site only) | | | ✔ | | ✔ |
| Physical functioning | SPPB[49] | | | ✔ | | ✔ |
| Muscle strength | Hand grip strength | | | ✔ | | ✔ |
| Muscle strength | Leg extension strength | | | ✔ | | ✔ |
| Self-reported mobility limitations | Ability to walk 400 m and climb one flight of stairs | | | ✔ | ✔ | ✔ |
| Risk of sarcopenia | SARC-F Questionnaire[66] | | | ✔ | ✔ | ✔ |
| Malnutrition | BMI <22.0 kg/m$^2$ or unintentional weight loss >5% in the last 6 months | | | ✔ | | ✔ |
| Frailty | Frailty Fried Frailty Index[64] | | | ✔ | | ✔ |
| Quality of Life | EuroQol 5D[63] | | | ✔ | ✔ | ✔ |
| Healthcare costs | RUD[67] | | | ✔ | ✔ | ✔ |
| Appetite | SNAQ-Appetite[68] | | | ✔ | | ✔ |
| Physical activity | Accelerometers | | | ✔ | ✔ | ✔ |
| Process evaluation | | | | | | ✔ |
| *Persuasive technology study* | | | | | | |
| Communication style preferences | Personality traits form | | | ✔ | | |
| Usage data of technology | Practical experiences, interaction data of technology (number of notifications, openings, registered food intake, games) | | | | ✔ | ✔ |
| Attitude towards technology | Questionnaire | | | | | ✔ |
| *Microbiota study* | | | | | | |
| Oral health | Questionnaire | | | ✔ | | ✔ |
| Oral microbiota | Tongue swab (16S rRNA sequencing) | | | ✔ | | ✔ |
| Gut microbiota | Fresh frozen faecal sample (16S rRNA sequencing) | | | ✔ | | ✔ |

**Table 1** Continued

| | | Timeline | | | |
|---|---|---|---|---|---|
| | | Screening (phone) | Screening visit (prior baseline) | Baseline visit | 3-month FU visit | 6-month FU visit |
| *fMRI study* | | | | | | |
| Oral microbiota | Fasted unstimulated salivary sample (16S rRNA sequencing) | | | ✔ | | ✔ |
| Nutritional and microbial markers | Blood sample | | | ✔ | | ✔ |
| Appetite | VAS scores of appetite and central neural responses to food-cues measured by fMRI scan | | | ✔ | | ✔ |

BMI, body mass index; fMRI, functional MRI; FU, follow-up; MMSE, Mini-Mental State Examination; rRNA, ribosomal RNA; RUD, Resource Utilisation in Dementia; SNAQ, Simplified Nutritional Appetite Questionnaire; SPPB, Short Physical Performance Battery; VAS, Visual Analogue Scale.

way. Furthermore, identical participant brochures and other printed materials have been developed and translated to Dutch and Finnish language.

## Outcomes
### Primary outcome
The primary outcome of the PROMISS trial is a 6-month change in physical functioning measured by change in walk time using a 400 m walk test (Long Distance Corridor Walk).[44 45] This test is predictive for higher risk of mortality, incident cardiovascular disease, and mobility limitation and disability.[46] One advantage of this continuous outcome is that it enables discrimination between categories of risk among participants,[47] and it is less prone to a ceiling effect as compared with other functional outcome measures (eg, SPPB).[48] The course for the 400 m test is 20 m long and marked by a traffic cone and tape line at the beginning and end. For all participants, the test will begin with a mandatory 40 m walk (warm-up) at their usual pace. Thereafter, the 400 m test starts with the feet behind and just touching the starting line and ends after 10 complete rounds when one foot is behind the end line. For the 400 m test, older adults will be instructed to walk as fast as possible at a pace they can maintain for 400 m. Standardised encouragement will be given each lap, including the number of laps remaining. At the 6-month follow-up visit, older adults are allowed to use a cane, can take a rest as needed (but no rest longer than 60 s) and there will be a time limit of 17 min. Time will be recorded to the nearest second.

### Secondary outcomes
The secondary outcomes are changes in physical performance, muscle strength, body composition, self-reported mobility limitations, QoL, incidence of frailty, incidence of sarcopenia risk and incidence of malnutrition. We will also investigate change in healthcare costs.

Physical performance will be assessed by means of the SPPB.[49] The SPPB assesses lower extremity function which consists of three timed tests: repeated (five times) chair stand test, 4-metre walk test and three standing balance tests (ability to stand with the feet together in the side-by-side, semitandem and tandem positions). The total score ranges from 0 to 12. A higher score indicates better physical functioning.

Muscle strength will be determined by hand grip strength (kg). Hand grip strength is an indicator of overall muscle strength[50] and a higher hand grip strength is associated with decreased risk of physical disabilities[51] and all-cause mortality in old age.[51 52] Maximum grip strength will be measured three times at each hand during baseline and 6-month follow-up visit. We will use a digital dynamometer (Saehan Digital Handheld Dynamometer) adjusted for hand size. Participants will be measured in an upright sitting position with the forearms supported by the armrest of a chair according to a standardised protocol.[53] The mean of the maximum score of left and right hand will be used for analyses. Muscle strength will also be determined by leg extension strength (N). A higher leg extension strength is associated with decreased risk of mobility disability[54 55] and lower risk of early mortality.[56–58] Leg extension strength will be assessed using a chair designed to measure leg extension strength.[59] Maximum leg extension strength will be measured three times for each leg during baseline and 6-month follow-up visit. The mean of the maximum score of left and right leg will be used for analyses.

Body composition will be estimated by means of bioelectrical impedance using the BodyStat 1500MDD device, using the Kyle equation to determine fat percentage (%), fat mass and fat-free mass[60] and the Sergi equation to determine appendicular skeletal muscle mass (kg).[61] Additionally, at the Dutch study site, body composition (fat free mass (kg), fat mass and fat percentage (%)) will be measured by air displacement plethysmography.[62]

Self-reported mobility limitations will be assessed by means of a questionnaire; 'Because of your health, how much difficulty do you have walking 400 metre?' and 'Because of your health, how much difficulty do you have climbing 10 steps?' Participants will respond using a 5-level Likert scale: 'no difficulty', 'a little difficulty', 'some difficulty', 'a lot of difficulty' and 'unable to do the activity'. Mobility limitation is defined as two consecutive

reports of having any difficulty walking 400 m or climbing 10 steps without resting due to a health or a physical problem.

QoL will be measured using the EuroQol 5D-5L Questionnaire.[63]

Incident frailty will be assessed using the Fried criteria.[64] Participants will be considered 'frail' when three or more components are present. Those with no components will be considered 'robust', whereas those with one or two components will be considered 'prefrail'. The criteria include (1) self-reported unintentional weight loss (>4 kg in past 6 months), (2) self-reported exhaustion (based on two questions from the Center for Epidemiologic Studies Depression Scale on exhaustion in the past week at baseline and follow-up: 'I felt that everything I did was an effort' and 'I could not get going'. Scores range from 1 'rarely or none of the time' to 4 'always or most of the time'. A score of 3 or 4 on either question indicates exhaustion,[65] (3) weakness (grip strength in the lowest 20% of the whole study population based on the mean of the maximum scores, adjusted for gender and BMI), (4) slow walking speed (walk time on the 4 m walk test in the slowest 20% of the whole study population, adjusted for gender and height), and (5) low physical activity (total counts per week based on the accelerometer data in the lowest 20% of physical activity for each gender). Incident frailty is considered deterioration of frailty status; that is, from robust at baseline to prefrail or frail at follow-up or from prefrail at baseline to frail at follow. Frail participants at baseline will be excluded from these analyses.

Incidence of sarcopenia risk will be assessed with the SARC-F Questionnaire[66]; how much effort do you experience when (1) lifting and carrying a bag of 4.5 kilo, (2) walking across a room, (3) transferring from a chair or bed, (4) climbing a flight of 10 stairs and (5) how many times have you fallen in the past year. Answering option include no effort (0 points), a bit of effort (1 point) and a lot of effort (2 points), where a score equal to or greater than 4 is predictive of sarcopenia and poor outcomes. Participants with risk of sarcopenia at baseline will be excluded from these analyses.

Incidence of malnutrition will be defined as BMI <22.0 kg/m$^2$ or unintentional weight loss >5% in the last 6 months. Malnourished participants at baseline will be excluded from these analyses.

A modified version of the Resource Utilisation in Dementia Questionnaire[67] will be used to collect data on healthcare and social utilisation costs over the period 3 months prior to baseline, 3 months prior to the 3-month follow-up visit and 3 months prior to 6-month follow-up visit. Costs include costs of primary and secondary care, complementary care, informal care and home care.

## Other measures
BW will be measured without shoes in underwear to the nearest 0.1 kg using a digital calibrated scale (Finland; SECA 877, the Netherlands; Marsden M-520). Body height will be measured to the nearest millimetre using a SECA

stadiometer for mobile height measurements (Finland; SECA 217, the Netherlands; SECA 214). Corrections will be made to adjust the measured BW for clothing, shoes or a cast (minus 1 kg for each element), and to adjust the measured body height for shoes (minus 1 cm). Change in BW and BMI at 3 months and 6 months will be calculated. As the personalised dietary advice is isocaloric, no significant difference in BW or BMI is expected. Physical activity will be objectively assessed by means of an accelerometer (Axivity, AX3) during 7 subsequent days after each clinic visit (baseline, 3-month follow-up visit and 6-month follow-up visit). The accelerometer will be attached by a nutritionist to the frontal part of the right thigh in the midpoint between iliac crest and patella bone when sitting down, with a surgical plaster. Participants can perform any physical activity as the accelerometer is water resistant. Appetite will be measured with Simplified Nutritional Appetite Questionnaire.[68] Dietary intake will be assessed by means of a combination of three food diaries and three 24-hour dietary recalls prior to the 3-month follow-up visit and the 6-month follow-up visit.

## Sample size and statistical analyses
### Sample size
The study is powered to detect a substantial meaningful change of 28 s (SD=61 s)[69] between the respective intervention groups and the control group on the primary outcome walk time on the 400 m walk test, assuming a two-sided test at α=0.05 with a power of 0.8. For this, 75 participants per group are needed, which is 225 in total. Assuming a drop-out of 15% (which was reported in a comparable study of Bhasin et al,[19] the total number of study participants to be included in the study is n=264. Therefore, we aim to include a total of n=132 at each study site (n=44 per study group per study site).

### Statistical analyses plan
Statistical reporting will be according to the Consolidated Standards of Reporting Trials standards.[70] The collected data at the two study sites will be pooled together at the Amsterdam site, with a variable indicating study site. All statistical analyses on primary and secondary outcome measures will be performed by an independent statistician blinded for group allocation. Baseline characteristics will be described (percentages, means±SDs) by study group.

The primary analyses will be based on the intention-to-treat principle, that is, data from participants allocated to the intervention groups will be analysed as part of those groups, irrespectively of their level of adherence to the advice. Multiple imputation using Multivariate Imputation by Chained Equations will be used to impute missing cost and effect data. The continuous primary outcome (change in 400 m walk time) will be analysed using mixed model regression analyses with study site as random variable. We will adjust for baseline 400 m walk time as well as baseline protein intake (g/kg aBW/day) and sex (the stratification factors for randomisation). We will compare

intervention effects of the respective intervention groups versus the control group. Effects will also be expressed in Cohen's d and the corresponding 95% CIs will be calculated, which allows comparison between intervention effect estimates between different outcome measures. The secondary outcomes and other measures will be analysed using mixed model regression analogously to the primary outcome. For binary secondary outcomes, generalised estimating equation models will be used. With regard to time-to-event analyses (incident sarcopenia risk and mobility limitations) Cox proportional hazard models will be used. Time-to-event is defined as the time of the start of the study period to the date of the first occurrence of the event (3-month follow-up visit or 6-month follow-up visit). Participants who do not meet these criteria will be censored at the latest time we had information available. We will perform subgroup analyses stratified by baseline protein intake (<0.9 or 0.9–1.0 g/kg aBW/day), sex and baseline 400 m time (based on median) for the primary and secondary outcomes.

Per-protocol analyses will also be conducted as a sensitivity analysis. Effect estimates for change in primary and secondary outcome measures will be calculated for participants from the intervention groups who reached the protein target of at least 1.2 g/kg aBW/day after both 3 and after 6 months (mean protein intake based on the three 24-hour dietary recalls) versus participants from the control group. Data will be analysed using SPSS (IBM SPSS Statistics, Version 26). A two-sided p value of 0.05 is considered statistically significant.

The cost-effectiveness analysis will be performed from a healthcare perspective. Total mean costs during the study will be related to physical functioning (the 400 m walk test) and change in quality-adjusted life-years based on the EuroQol 5D Questionnaire. Mixed model regression analyses will be used to estimate differences in the primary outcome of the respective intervention groups versus the control group. Linear regression analyses will be used to estimate differences in QoL (expressed as quality-adjusted life-years) and healthcare costs. Incremental cost-effectiveness ratios will be calculated by dividing the difference in costs by the difference in effects. Statistical uncertainty will be estimated using bias-corrected accelerated bootstrapping (5000 replications) and will be presented using cost-effectiveness planes and cost-effectiveness acceptability curves.

## Participants' safety
In case any (medical) questions arise during the screening or intervention period, participants can consult an independent medical doctor. All adverse events and serious adverse advents will be tracked by the nutritionists during the follow-up phone calls, 3-month follow-up visit and 6-month follow-up visit to assess their potential relationship to the intervention at both sites and will be documented in the final report. Adverse events will be reported within 7 days (death or life-threatening situations) or within 15 days (in case of other adverse events)

of first knowledge to The Medical Ethical Committee of the Amsterdam UMC, location VUmc (required for the Dutch site only).

## Data quality assurance and data management
Research data will be collected at each site and each visit (baseline, 3-month follow-up visit and 6-month follow-up visit) using a standardised protocol with the same order of assessments, and entered two times in separate electronic datasets. When discrepancies between the datasets are found, the original questionnaire will be consulted. Questionnaire items and measurements will include the corresponding variable names to minimise errors in data entering. Finally, the two final electronic datasets (one from each site) containing all data will be pooled. A data catalogue and codebook will be developed.

Original questionnaires will be stored in a secure manner at each site in an area with limited access. All records that contain names (ie, informed consent forms) will be stored separately from study records identified by code number. All databases will be secured with password-protected access systems.

## Patient and public involvement
The PROMISS RCT is designed by the Faculty of Science (VU Amsterdam, the Netherlands) and the Department of General Practice and Primary Health Care (University of Helsinki, Finland) (please see online supplemental appendix 1 for the PROMISS trial group), a collaboration of the EU Horizon 2020 PROMISS Project. Two medical and one ethical advisor are involved in the study. As part of the PROMISS project, we previously performed three pilot studies as a preparation of the long-term PROMISS randomised trial, of which one is published.[71] We included the feedback of the participants in designing the long-term PROMISS randomised trial; participants enjoyed participating in the pilot studies and they liked the frequent contact with the nutritionist. We also tested which protein-enriched food products they preferred and included those products in the long-term PROMISS randomised trial which they liked the most. Older adults are not involved in recruitment of participants or conduct of the study. Results of this study will be disseminated to participants through sending them a lay abstract with the results and conclusions of the study. At the end of the study, each participant will receive a fact-sheet with personal results of dietary intake data, hand grip strength, body composition measures and BW. Participant burden of the pilot intervention was assessed using informal feedback from older adults participating in one of three pilot studies.

## Ancillary studies
Within the PROMISS trial, three ancillary studies will be conducted: (1) persuasive technology study, (2) microbiota study and (3) functional MRI (fMRI) study.

## Persuasive technology study

The primary aim of this study is to examine the effect of persuasive technology on adherence to the personalised dietary advice aiming at increasing protein intake to at least 1.2 g/kg aBW/day in a subsample of Dutch participants from intervention group 1 (n=24) and intervention group 2 (n=24), that is, the first 24 participants of intervention group 1 and the first 24 participants of intervention group 2 that consent to it (written informed consent will be signed).

Participants will be provided with a food storage box that registers which provided protein-enriched food products are taken out. The food box is used to store the protein-enriched food products provided by the research team. Participants will also receive a tablet that allows participants to register any consumed protein-enriched food products and supports them in finding alternative food products that contain a comparable amount of protein. For this, the system uses the personalised dietary advice as provided by the nutritionist and data from the storage box. The tablet application aims to stimulate adherence to the dietary advice by providing tailored and personalised messages. In addition, personality characteristics and communication style preferences that are determined via a questionnaire completed at baseline are used to tailor the style and tone of these messages.[72]

In addition to the personalised messages, half of the participants from intervention groups 1 and 2 who participate in the persuasive technology substudy will also receive a gamified version of the tablet application (n=12+n=12). In this version, participants can earn game points by registering their consumed protein-(en)rich(ed) food products and by playing mini-games about the protein content of foods (ie, guess the protein content, more-or-less protein). The distribution of receiving the gamified version versus standard version is quasi-randomised, where we will balance the group size.

At the consultation meeting, participants receive their food storage box and tablet. Both are fully configured, that is, they are loaded with their personal dietary advice. After the 6-month follow-up visit, participants will be asked to return the equipment and fill out questions on the feasibility and user experience of the provided persuasive technology.

The secondary objectives are (1) to investigate to what extent participants perceive messages of which the style and tone are adapted to their personal characteristics as personalised and adequate, and (2) to determine the effect of gamification on the effectiveness and feasibility of the persuasive technology.

## Microbiota study

In the microbiota study, the effect of personalised dietary advice aiming at increasing protein intake in community-dwelling older adults with lower habitual protein intake on both the oral and gut microbiota is investigated. The study will be conducted at both study sites.

The human microbiota consists of the $4*10^{13}$ microorganisms that inhabit the body.[73] The emergence of next generation DNA sequencing techniques at the start of the 21st century has allowed more detailed study of the microbiota and since then, the microbiota composition has been associated with both health and disease,[74] as well as ageing itself.[75 76] Moreover, several interventional studies proved that dietary changes also affect the gut microbiota, with the first microbial shifts being evident within 48 hours.[77] The altered microbiota in turn, can differentially affect the human host metabolism through the production of metabolically active metabolites. Less is known about the oral microbiota. It was found to be associated with oral health and function and even nutritional status,[78 79] but its possible role in undernutrition in older adults has not been investigated.

A fresh frozen faecal sample and tongue swab are collected at baseline and 6-month follow-up visit once written informed consent is provided. Participants from either the control group or intervention group 1 can be included in this study. Participants from the intervention group 2 are excluded to limit the number of groups and parameters in this exploratory study. Additional exclusion criteria are: use of systemic antibiotics in the 3 months prior to the first sampling visit, diagnosis with inflammatory bowel disease and prolonged institutionalisation (>4 weeks) in the 3 months prior to the first sampling visit. There is no restriction other than consent rate to the number of PROMISS participants that will be included in this side study.

Once all samples from all participants are collected, faecal samples are shipped to the Wallenberg Laboratory of Cardiovascular and Metabolic research (at the University of Gothenburg, in Sweden) for 16S ribosomal RNA (rRNA) sequencing using sequencing methods previously described.[80] The tongue swabs will be sent to the Netherlands Organisation for Applied Scientific Research for 16S rRNA sequencing as is previously described.[81]

## fMRI study

In the fMRI study, we will investigate the effect of personalised dietary advice aiming at increasing protein intake in community-dwelling older adults with lower habitual protein intake on central brain circuits involved in the regulation of appetite. Several studies demonstrated that increasing protein intake affects appetite[82] and the gut microbiota.[83] However, none have studied the effects on both simultaneously, or the interaction. An fMRI scan will be used to measure the brain responses to visual or actual food cues. Brain activity in response to food cues will also be related to (shifts) in the gut microbiota. Therefore, only participants from the microbiota side study can be included in this study, with additional exclusion criteria: being claustrophobic, being diagnosed with a mental disorder (eg, depression or addiction), being uncorrectable visually or hearing impaired, or having a contra-indication for MRI scans (eg, having a pacemaker). Up

to 50 participants will be included in this side study. This side study will only be conducted at the Dutch study site.

Once written informed consent is provided, participants who are included in this side study will be asked to visit the Amsterdam University Medical Centre, location VUmc, for an fMRI scan two times during the study period: at baseline and at 6-month follow-up visit. Prior to the fMRI scan, additional salivary and blood samples will be collected for determination of additional nutritional and microbial biomarkers. The protocol for the fMRI experiments has been previously described.[84 85]

## DISCUSSION

There is an ongoing discussion whether the EFSA RDA of 0.8 g protein/kg BW/day is sufficient for older adults and whether it should be increased to at least 1.0–1.2 g protein/kg BW/day to support muscle health and functioning. National guidelines of some European countries already increased their RDA, that is, the RDA of the German-speaking countries (D-A-CH) is increased to 1.0 g/kg BW/day,[86] and the Nordic Nutrition Recommendation has increased their RDA to 1.2 g/kg BW/day.[87] The PROMISS trial is the first RCT which will investigate the effect of personalised dietary advice aiming at increasing protein intake and the combined effect of personalised dietary advice aiming at increasing protein and the timing of protein intake in close proximity of usual physical activity, on change in physical functioning after 6 months among community-dwelling older adults (≥65 years) with a habitual protein intake of <1.0 g/kg aBW/day. The PROMISS trial will therefore provide additional insight to the question whether the current EFSA RDA for protein for older adults should be increased to 1.2 g/kg aBW/day, and whether optimal timing of protein intake will additionally benefit physical functioning.

A strong and unique aspect of the PROMISS trial is that we will include participants with a habitual protein intake <1.0 g/kg aBW/day, excluding those with a BMI <18.5 and >32.0 kg/m$^2$. This will allow us to examine the effects of increasing protein intake from <1.0 to at least 1.2 g/kg aBW/day. An innovative component of our study is that we will investigate the combined benefit of increasing protein intake and timing of protein intake with usual physical activity on physical functioning and other health-related outcomes. Another strength is that in our study the intervention is based on personalised dietary advice which is likely more feasible in the long term to maintain in everyday life, compared with providing custom-prepared meals[19] or protein supplements,[88 89] as done in most other studies. Finally, we will be able to investigate the effect of persuasive technology on adherence to the dietary advice strategy, and the effect of the dietary advice on the microbiota composition and on central responses to food-cues in brain areas involved in appetite regulation. One limitation of this study is that the biological value of the total protein intake (ie, type of amino acids) is unknown. Another limitation is that the duration of the trial might not be long enough to observe a sufficient amount of incident cases of for example, risk of malnutrition, frailty or risk of sarcopenia.

In summary, this RCT will demonstrate the effectiveness of personalised dietary advice aiming at increasing protein intake to at least 1.2 g protein/kg BW/day on physical functioning in older adults with a lower habitual protein intake, with or without the advice to consume protein in close proximity of usual physical activity.

## ETHICS AND DISSEMINATION

The study has been approved by the Ethics Committee of the Helsinki University Central Hospital, Finland (ID of the approval: HUS/1530/2018) and The Medical Ethical Committee of the Amsterdam UMC, location VUmc, Amsterdam, the Netherlands (ID of the approval: 2018.399). Oral informed consent will be obtained from each participant before the screening procedure and written informed (please see online supplemental appendix 2) consent will be obtained from each participant before any measurement takes place. Personal data were not identifiable during the analysis.

Results will be sent to national and international conferences and will submitted for publication in peer-reviewed journals. In addition, lay abstracts will be made available for participants and the public. Links to research output and dissemination activities will be made available on the PROMISS website, available at www.promiss-vu.eu and social media channels.

**Author affiliations**

[1]Department of Health Sciences, Faculty of Science, and the Amsterdam Public Health research institute, Vrije Universiteit Amsterdam, Amsterdam, The Netherlands
[2]Department of General Practice and Primary Health Care, University of Helsinki, Helsinki, Finland
[3]Unit of Primary Health Care, Helsinki University Central Hospital, Helsinki, Finland
[4]Department of Internal Medicine, Amsterdam UMC, Vrije Universiteit Amsterdam, Amsterdam Public Health research institute, Amsterdam UMC, Amsterdam, The Netherlands
[5]Wallenburg Laboratory, Department of Molecular and Clinical Medicine, Sahlgrenska Academy, University of Gothenburg, Gothenburg, Sweden
[6]Department of Computer Science, Vrije Universiteit Amsterdam, Amsterdam, The Netherlands

**Acknowledgements** We acknowledge the members of the PROMISS trial group and we thank the study participants. We thank Jan de Vries for his ethical advice, and Martin den Heijer and Kaisu Pitkälä for their medical advice.

**Contributors** IR, HAHW, IAB, MRO and MV obtained funding for the PROMISS project. IR and HAHW coordinated the trial centre at the Vrije Universiteit Amsterdam, the Netherlands. SKJ and MHS coordinated the trial centre at the Helsinki University, Finland. All authors contributed to the conception and designing of the trial. IR drafted the manuscript. JEB provided cost-effectiveness expertise in clinical trial design. LDK provided statistical expertise and will conduct the primary statistical analysis. KSF drafted the sections for the microbiota and fMRI ancillary studies. MCAK and LMvdL drafted the section for the persuasive technology study. HAHW, SKJ, MHS, RN, IAB, MRO, KHP, RV and MV critically reviewed the manuscript. All authors approved the final version.

**Funding** Funding for this research is provided by EU Horizon 2020 PROMISS Project 'Prevention Of Malnutrition In Senior Subjects in the EU', Grant agreement

no. 678 732. Protein-enriched food products are provided by Kellogg and Fonterra. Costs for these products are also funded through the EU Horizon 2020 PROMISS grant.

**Competing interests**  None declared.

**Patient consent for publication**  Not required.

**Provenance and peer review**  Not commissioned; externally peer reviewed.

**ORCID iD**
Ilse Reinders http://orcid.org/0000-0003-2156-4254

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
