## [Reviewer comments · BMJ Open]

ARTICLE DETAILS

TITLE (PROVISIONAL)	Effectiveness and cost-effectiveness of personalized dietary advice aiming at increasing protein intake on physical functioning in community-dwelling older adults with lower habitual protein intake: rationale and design of the PROMISS randomized controlled trial
AUTHORS	Reinders, Ilse; Wijnhoven, Hanneke; Jyvakorpi, S; Suominen, Merja; Niskanen, Riikka; Bosmans, J; Brouwer, Ingeborg; Fluitman, Kristien; Klein, Michel; Kuijper, Lothar; van der Lubbe, Laura; Olthof, Margreet R.; Pitkala, Kaisu H.; van der Pols-Vijlbrief, Rachel; Visser, Marjolein

VERSION 1 – REVIEW

REVIEWER	Esther Lopez-Garcia Universidad Autónoma de Madrid
REVIEW RETURNED	14-Jul-2020

GENERAL COMMENTS	BMJOpen_20202-040637 This is a European funded RCT that addresses a key question for adequate nutrition in the older population: whether protein intake above the current recommended dietary allowance (RDA) of 0.8 g/protein/kg of body weight/day (specifically >1.2) is able to improve physical function and several measures related to musculoskeletal health. Additionally, this study also will examine whether this amount of protein ingested half an hour after physical activity provides improved benefits than random consumption during the day. This RCT build on the current literature that supports that a deficient intake of total proteins has a detrimental effect on sarcopenia (1), whereas intake above RDA has been shown to reduce hip fractures (2) and bone mass density loss (3), and helps maintain physical function (4,5). Although energy requirement declines with age because a reduction in the basal metabolic rate (6), the need for protein intake increases in order to compensate for age-related decreases in skeletal muscle mass, strength, and function. The current RDA for protein is based on short-term
---

nitrogen balance studies, which may not be the best methods to estimate long-term habitual requirements for the older adults (7). More recent recommendations propose an average daily intake of 1.0-1.2 g/kg/day for older adults and even higher for those with acute or chronic diseases (8). In summary, although the current literature suggests that older adults are at increased need for high quality protein, it is not well established the amount of protein intake that prevents physical function impairment and derived outcomes, such as frailty, disability and poor quality of life. This RCT sounds relevant and timely.

Some strengths of this protocol includes an appropriate background and detailed information of the main and ancillary outcomes. However, the protocol would benefit of additional information:

1. It is unclear the meaning of personalized dietary advice. Does it mean that the researchers are going to take into account additional information than the macronutrient intake? This seems reasonable. The reviewer thinks about the ability to cook, whether they live alone or they have help in daily tasks including cooking, habitual dietary habits (one main meal a day vs. small and frequent eating occasions), food preferences, and also dietary restrictions due to medical problems.
2. The protocol explains that food will be provided to the participants. This strategy has also been successfully used in the PREDIMED study. However, there is no specification of which types of foods are going to be included. This is key to understand the external validity of the intervention and should be included in the protocol of the RCT.
3. Proteins from different sources vary in amino acid profiles, which may have different effects on muscle protein synthesis. Specifically, 'fast' proteins such as whey and soy protein are rapidly digested and absorbed and may therefore have a great impact on muscle protein accretion. Or leucine, mainly present in animal products is suggested to have a positive effect on signaling pathways for muscle

protein synthesis. Previous research has found that increased intake of vegetable protein, but not animal protein, has been associated with delayed unhealthy aging (10). Which type of proteins are going to be used in this intervention? At this point of the current knowledge, this information is critical.

4. Some evidence in the area of chronobiology suggests that an even protein intake during the day has more benefit on health than an uneven intake. The researchers consider that participants will have to comply with the requirement of consumption of at least one daily meal >35 g protein. Do they plan to address the impact of an even vs. an uneven diet? The coefficient of variability in protein intake is a good method to deal with this.
5. Several of the secondary endpoints referenced may take more than 6 months to develop. In fact, longitudinal studies assessing habitual diet seem more appropriate than a RCT. Have the authors anticipated that they may not observe incident cases of malnutrition, or frailty, or sarcopenia in this period of time?
6. Finally, I am curious about the attendance to non-health lectures in this older population. Do the authors have experience on the acceptability of this program? The participants need a similar (and high) socioeconomic level and specific sociodemographic characteristics to be attracted to the lectures. More information about this can be useful not only to confirm that lectures will be useful but also for a better understanding of population characteristics, in order to have a clue for the extrapolation of the intervention.

References

1. Cruz-Jentoft AJ, Dawson Hughes B, Scott D, Sanders KM, Rizzoli R. Nutritional strategies for maintaining muscle mass and strength from middle age to later life: A narrative review. *Maturitas*. 2020;132:57-64.
2. Fung TT, Meyer HE, Willett WC, Feskanich D. Protein intake and risk of hip fractures in postmenopausal women and men age 50 and older. *Osteoporos Int* 2017;28(4):1401–11.
3. Rizzoli R, Biver E, Bonjour JP, Coxam V, Goltzman D, Kanis JA, et al. Benefits and safety of dietary protein for bone health-an expert consensus paper endorsed by the European Society for Clinical and Economical Aspects of Osteoporosis, Osteoarthritis, and Musculoskeletal Diseases and by the International Osteoporosis Foundation. *Osteoporos Int*. 2018;29(9):1933–1948.

	 4. Rizzoli R, Stevenson JC, Bauer JM, et al. The role of dietary protein and vitamin D in maintaining musculoskeletal health in postmenopausal women: a consensus statement from the European Society for Clinical and Economic Aspects of Osteoporosis and Osteoarthritis (ESCEO). Maturitas 2014;79:122–32. 5. Lisset E M Elstgeest, Laura A Schaap, Martijn W Heymans, Linda M Hengeveld, Elke Naumann, Denise K Houston, Stephen B Kritchevsky, Eleanor M Simonsick, Anne B Newman, Samaneh Farsijani, Marjolein Visser, Hanneke A H Wijnhoven, for the Health ABC Study. Sex-and race-specific associations of protein intake with change in muscle mass and physical function in older adults: the Health, Aging, and Body Composition (Health ABC) Study. Am J Clin Nutr 2020;112:84–95. 6. Institute of Medicine. Energy. In: Dietary Reference Intakes for Energy, Carbohydrate, Fiber, Fat, Fatty Acids, Cholesterol, Protein, and Amino Acids, Washington, DC: National Academies Press; 2005.p. 143. 7. Volpi E, Campbell WW, Dwyer JT, et al. Is the optimal level of protein intake for older adults greater than the recommended dietary allowance? J Gerontol A Biol Sci Med Sci 2013;68:677–81. 8. Bauer J, Biolo G, Cederholm T, et al. Evidence-based recommendations for optimal dietary protein intake in older people: A position paper from the PROT-AGE Study Group. J Am Med Dir Assoc 2013;14(8):542–59 9. Ortolá R, Struijk EA, García-Esquinas E, Rodríguez-Artalejo F, Lopez-Garcia E. Changes in dietary intake of animal and vegetable protein and unhealthy aging. Am J Med. 2020;133(2):231–239.
--	---

REVIEWER	Olof Gudny Geirsdottir University of Iceland
REVIEW RETURNED	16-Aug-2020

GENERAL COMMENTS	Important study for all recommendation about food and diet for elderly. Eligibility criteria is BMI between 18.5-32, BMI under 18.5 is understandable because of risk of malnutrition. However, BMI 32 is rather low in the light of mean BMI of old EU adults is about BMI 28, and recommendation about dietary restrictions are debatable and need reference.
---

VERSION 1 – AUTHOR RESPONSE

Reviewer(s) Reports:

Reviewer: 1

Reviewer Name: Esther Lopez-Garcia

Institution and Country: Universidad Autónoma de Madrid Competing interests: None declared

This is a European funded RCT that addresses a key question for adequate nutrition in the older population: whether protein intake above the current recommended dietary allowance (RDA) of 0.8 g/protein/kg of body weight/day (specifically >1.2) is able to improve physical function and several measures related to musculoskeletal health. Additionally, this study also will examine whether this amount of protein ingested half an hour after physical activity provides improved benefits than random consumption during the day.

This RCT build on the current literature that supports that a deficient intake of total proteins has a detrimental effect on sarcopenia (1), whereas intake above RDA has been shown to reduce hip fractures (2) and bone mass density loss (3), and helps maintain physical function (4,5). Although energy requirement declines with age because a reduction in the basal metabolic rate (6), the need for protein intake increases in order to compensate for age-related decreases in skeletal muscle mass, strength, and function. The current RDA for protein is based on short-term nitrogen balance studies, which may not be the best methods to estimate long-term habitual requirements for the older adults (7). More recent recommendations propose an average daily intake of 1.0-1.2 g/kg/day for older adults and even higher for those with acute or chronic diseases (8). In summary, although the current literature suggests that older adults are at increased need for high quality protein, it is not well established the amount of protein intake that prevents physical function impairment and derived outcomes, such as frailty, disability and poor quality of life. This RCT sounds relevant and timely.

Some strengths of this protocol includes an appropriate background and detailed information of the main and ancillary outcomes. However, the protocol would benefit of additional information:

1. It is unclear the meaning of personalized dietary advice. Does it mean that the researchers are going to take into account additional information than the macronutrient intake? This seems reasonable. The reviewer thinks about the ability to cook, whether they live alone or they have help in daily tasks including cooking, habitual dietary habits (one main meal a day vs. small and frequent eating occasions), food preferences, and also dietary restrictions due to medical problems.

Based on dietary intake data of three 24-h recalls, we have information on participants' dietary habits beyond macronutrient intake. The aim of the dietary advice is to increase protein intake to at least 1.2 g/kg aBW/d. This is only feasible on the long term when the dietary changes which participants need to make fit within current dietary habits. Therefore, we indeed based the personalized advice on participants' current habits. We have also asked participants if they are the one who usually prepares the main meal; whether they eat the meal at a e.g. community home; whether they consume ready-to-eat meals; whether they use meal services; and if they eat at family or friends' home. All their answers are incorporated when the nutritionist composed the dietary advice.

We have added this information to the *Intervention* section of the manuscript.

2. The protocol explains that food will be provided to the participants. This strategy has also been successfully used in the PREDIMED study. However, there is no specification of which types of foods are going to be included. This is key to understand the external validity of the intervention and should be included in the protocol of the RCT.

The protein enriched food products used are protein bars, cereals, puddings, coconut water and whey powder, which will be freely provided and shipped to participants' home (stated under the section Intervention).

3. Proteins from different sources vary in amino acid profiles, which may have different effects on muscle protein synthesis. Specifically, 'fast' proteins such as whey and soy protein are rapidly digested and absorbed and may therefore have a great impact on muscle protein accretion. Or leucine, mainly present in animal products is suggested to have a positive effect on signaling pathways for muscle protein synthesis. Previous research has found that increased intake of vegetable protein, but not animal protein, has been associated with delayed unhealthy aging (10). Which type of proteins are going to be used in this intervention? At this point of the current knowledge, this information is critical.

We thank the reviewer for this comment. The protein enriched food products we provide are of high quality (whey protein powder, cocowhey protein drink). It is however possible that they only eat a little of these products and the desired increase in protein intake is achieved by regular protein rich food

products of which the amino acid quality might be lower. This is indeed a limitation and have now acknowledged this limitation in the 'strength and limitations of this study' and 'discussion' section.

4. Some evidence in the area of chronobiology suggests that an even protein intake during the day has more benefit on health than an uneven intake. The researchers consider that participants will have to comply with the requirement of consumption of at least one daily meal >35 g protein. Do they plan to address the impact of an even vs. an uneven diet? The coefficient of variability in protein intake is a good method to deal with this.

As part of the preparation of the long term intervention trial, we investigated the feasibility of two dietary advice strategies to increase protein intake following either an even distribution of protein over the day ('even' strategy) or a peak in protein during one meal moment ('peak' strategy) (1). The results of that pilot study showed that both the 'even' and 'peak' dietary strategy were effective in substantially increasing protein intake in four weeks. In addition, participants following the 'peak' strategy more often had at least one meal per day with very high in protein. The knowledge from this pilot study, including the knowledge that consuming > 35 g of protein increased MPS in older adults (2-4) as mentioned in our paper, made us decide to advice participants to consume one meal high in protein.

We do not plan to investigate the effect of increasing total protein intake by means of a 'even' strategy, as this is also not possible since participants were advice to consume one meal very high in protein, however we are very interested when other research groups do and will keep following the new developments.

5. Several of the secondary endpoints referenced may take more than 6 months to develop. In fact, longitudinal studies assessing habitual diet seem more appropriate than a RCT. Have the authors anticipated that they may not observe incident cases of malnutrition, or frailty, or sarcopenia in this period of time?

The reviewer is correct. We acknowledge that our trial might not have included enough participants in order to have a sufficient amount of incident cases of one secondary outcome, or to have enough power to detect a meaningful difference on other secondary outcomes. We have now acknowledged this in the discussion.

6. Finally, I am curious about the attendance to non-health lectures in this older population. Do the authors have experience on the acceptability of this program? The participants need a similar (and high) socioeconomic level and specific sociodemographic characteristics to be attracted to the lectures. More information about this can be useful not only to confirm that lectures will be useful but also for a better understanding of population characteristics, in order to have a clue for the extrapolation of the intervention.

The main aim of the lectures was to increase involvement in the trial. Therefore we did not note attendance and do not know whether participants who attended the lectures had a higher SES compared to those who did not attend. The topics of the non-health related lectures were beekeeping and honey, and non-food sustainability (the Netherlands), and oral health and trusteeship (Finland). Participants could freely attend those lectures and all travel costs were reimbursed. Therefore, there were no financial barriers.

We have expanded this part in the manuscript.

Reviewer: 2

Reviewer Name: Olof Gudny Geirsdottir

Institution and Country: University of Iceland Competing interests: None declared

Please leave your comments for the authors below Important study for all recommendation about food and diet for elderly. Eligibility criteria is BMI between 18.5-32, BMI under 18.5 is understandable because of risk of malnutrition. However, BMI 32 is rather low in the light of mean BMI of old EU adults is about BMI 28, and recommendation about dietary restrictions are debatable and need reference.

We thank the reviewer for this interesting comment. Indeed, mean BMI of older adults is relatively high. However, a higher BMI ($> 30 \text{ kg/m}^2$) in older adults is associated with poorer physical function (5) and disability (6) and intentional weight loss by lifestyle interventions lead to a reduced mortality risk (7). Although we do not want to include extreme obese participants, we have chosen for the cut-off of 32 kg/m^2 to ensure we have a larger range in BMI. We have therefore chosen to exclude those people with a BMI of $> 32.0 \text{ kg/m}^2$.

We have added references indicating that a high BMI (>30 kg/m²) is associated with poorer physical function and disability.

FORMATTING AMENDMENTS (if any)

Required amendments will be listed here; please include these changes in your revised version:

1. Research Ethics number/ID of the Approval:

- You have indicated 'Yes' to this question. With this, please indicate the number/ID of the approval(s).

We have added the numbers of the approved research proposal.

2. Required figure/s format:

- Figures can be supplied in TIFF, JPG or PDF format (figures in document, excel or powerpoint format will not be accepted), we also request that they have a resolution of at least 300 dpi and 90mm x 90mm of width. Please see the following link for further details on preparing images for submission:

<https://authors.bmj.com/writing-and-formatting/formatting-your-paper/>

Figure 1 is now uploaded in PDF format.

3. Required Supplementary format:

- Please re-upload your Supplementary files in PDF format.

Done.

4. Patient and Public Involvement:

- We have implemented an additional requirement to all articles to include 'Patient and Public Involvement' statement within the main text of your main document. Please refer below for more information regarding this new instruction:

Authors must include a statement in the methods section of the manuscript under the sub-heading 'Patient and Public Involvement'.

This should provide a brief response to the following questions:

- **How was the development of the research question and outcome measures informed by patients' priorities, experience, and preferences?**
- **How did you involve patients in the design of this study?**
- **Were patients involved in the recruitment to and conduct of the study?**
- **How will the results be disseminated to study participants?**
- **For randomised controlled trials, was the burden of the intervention assessed by patients themselves?**

Patient advisers should also be thanked in the contributorship statement/acknowledgements.

If patients and or public were not involved please state this.

We have added this section, including the required information.

References

1. Reinders I, Visser M, Wijnhoven HAH. Two dietary advice strategies to increase protein intake among community-dwelling older adults: A feasibility study. *Clin Nutr ESPEN*. 2020;37:157-167.
2. English KL, Paddon-Jones D. Protecting muscle mass and function in older adults during bed rest. *Curr Opin Clin Nutr Metab Care*. 2010;13:34-39.
3. Paddon-Jones D, Rasmussen BB. Dietary protein recommendations and the prevention of sarcopenia. *Curr Opin Clin Nutr Metab Care*. 2009;12:86-90.
4. Deutz NE, Wolfe RR. Is there a maximal anabolic response to protein intake with a meal? *Clin Nutr*. 2013;32:309-313.
5. Kim S, Leng XI, Kritchevsky SB. Body Composition and Physical Function in Older Adults with Various Comorbidities. *Innov Aging*. 2017;1:igx008.
6. Vincent HK, Vincent KR, Lamb KM. Obesity and mobility disability in the older adult. *Obes Rev*. 2010;11:568-579.
7. Kritchevsky SB, Beavers KM, Miller ME, Shea MK, Houston DK, Kitzman DW, et al. Intentional weight loss and all-cause mortality: a meta-analysis of randomized clinical trials. *PLoS One*. 2015;10:e0121993.

VERSION 2 – REVIEW

REVIEWER	Olof Gudny Geirsdottir Faculty of Food and Nutrition, School of Health, University of Iceland
REVIEW RETURNED	19-Oct-2020
GENERAL COMMENTS	Interesting and important article in the field of nutrition. All comments has been answered by authers